# Impact of Enteral Nutrition on Clinical Outcomes in Very Low Birth Weight Infants in the NICU: A Single-Center Retrospective Cohort Study

**DOI:** 10.3390/nu17071138

**Published:** 2025-03-25

**Authors:** Pasqua Anna Quitadamo, Laura Comegna, Alessandra Zambianco, Giuseppina Palumbo, Maria Assunta Gentile, Antonio Mondelli

**Affiliations:** 1NICU, Casa Sollievo della Sofferenza Institute, 71013 San Giovanni Rotondo, Italy; l.comegna@operapadrepio.it (L.C.); a.mondelli@operapadrepio.it (A.M.); 2San Raffaele Faculty of Medicine, University of San Raffaele Vita-Salute, 20132 Milan, Italy

**Keywords:** human milk, human milk banking, donor milk, preterm infant feeding, breast milk, infant nutrition

## Abstract

**Background/Objectives**: Maternal milk feeding in the NICU (neonatal intensive care unit) for very low birth weight (VLBW) infants mitigates the effects of preterm birth. This single-center retrospective study analyzed data from VLBW infants born between 2005 and 2019 and investigated the impact on morbidity of exposure to Mother’s Own Milk (MOM), donor human milk (DHM), preterm formula (PF), during NICU hospitalization. The assessed outcomes included necrotizing enterocolitis (NEC), retinopathy of prematurity (ROP), bronchopulmonary dysplasia (BPD), and late-onset sepsis (LOS). The study also examined the impact of a human milk-based feeding protocol on these outcomes, adjusting for confounding factors. **Methods**: Statistical analysis involved correlation tests and odds ratios to assess associations between feeding types and outcomes. **Results**: Surgical NEC occurred in 10% of infants fed exclusively with PF, 1.3% of those fed with DHM, and was completely absent in infants fed exclusively or partially with MOM. ROP across all stages was observed in 24.3% of cases, with severe ROP at 4.7%, and PF feeding was associated with a higher risk of severe ROP; the incidence of LOS was lower in infants fed human milk (−22%/−66%) compared to 10% in formula-fed infants. BPD affected 25.5% of infants, with moderate-to-severe BPD in 22.2%. The association between NEC, LOS, and feeding was statistically significant, even after adjusting for covariates. The type of milk had a significant impact on the incidence of severe forms of all outcomes (*p* < 0.001). The rate of exclusive MOM feeding increased over time, reaching 45% in 2018–2019. **Conclusions**: These findings highlight the role of human milk in preventing NEC and LOS, in reducing the risk of severe ROP and BPD, and in promoting MOM feeding, with rates increasing significantly when DHM is available.

## 1. Introduction

The study of the benefits of maternal milk and its role in preventing neonatal and pediatric diseases represents a prominent area of research. Among preterm infants, increasing survival rates at progressively lower gestational ages, coupled with significant advances in biotechnology, have shifted the focus of Neonatal Intensive Care Units (NICUs) and research efforts over the past decade toward minimizing both short- and long-term morbidity. A central role of neonatal units is to support the growth and development of preterm infants during the critical period from birth to term-equivalent age.

Supported by national and international health organizations [1], evidence demonstrates that exclusive human milk-based feeding is one of the most effective strategies for reducing complications associated with preterm birth [2,3,4]. Its impact is particularly significant in preventing necrotizing enterocolitis (NEC) and late-onset sepsis (LOS), two of the most severe complications in this population [5,6,7]. Furthermore, numerous studies have shown that high-dose feedings of the mother’s own milk (MOM) during critical periods of NICU hospitalization reduce the incidence, severity, and risk of other potentially preventable morbidities, including retinopathy of prematurity (ROP) [8,9,10], bronchopulmonary dysplasia (BPD) [11,12,13], rehospitalization after NICU discharge, and neurodevelopmental delays in infancy and childhood [14,15,16].

Breast milk is widely regarded as the optimal nutrition for infants due to its species-specific composition, excellent tolerability, and, most importantly, its ability to protect the infant through various mechanisms. These include epigenetic modulation, stimulation of both innate and adaptive immunity, and promotion of physiological growth [1,17].

Although formula milk is designed to meet the high nutritional demands of preterm infants, it cannot replicate the complex composition of human milk (HM), which contains thousands of bioactive elements and multifunctional factors, many of which remain incompletely understood.

The first 2–3 weeks of life represent the most critical period for preterm infants in the NICU [18]. In very low birth weight infants (VLBWIs), the intestinal barrier is immature during the early postnatal period and exhibits an exaggerated immune and inflammatory response to exogenous stimuli, including nutritional and infectious challenges [19,20,21]. Breast milk promotes the rapid maturation of the intestinal barrier and mitigates its hyperreactivity [22,23], thereby protecting the infant from the development of necrotizing enterocolitis.

In this context, the use of donor human milk (DHM) allows for the avoidance of formula milk, which contains intact bovine milk proteins that can trigger an inflammatory response linked to the pathogenesis of NEC [24]. This approach is particularly valuable during the period before adequate maternal milk production is established. Although pasteurization reduces some bioactive components, DHM retains species-specific factors that are valuable for their biological functions and protective effects on the immature organism [5,6,25,26,27,28,29,30,31,32,33,34].

NEC affects approximately 5–10% of preterm infants weighing less than 1500 g [35], according to the NEC Society. However, at lower gestational ages, the incidence can rise to 22%, with mortality rates ranging from 21.9% to 38% and significant short- and long-term morbidity among survivors [36,37]. Data from the Vermont Oxford Network indicate that the prevalence of NEC in very low birth weight infants between 2007 and 2013 was 6.8%. In Italy, the prevalence of NEC during the same period was 4.3% [38], declining to 4% in 2021, with rates in some Italian NICUs ranging between 10% and 20%. The incidence is significantly higher in infants born at less than 28 weeks of gestation, reaching 9% [39]. Notably, 72.9% of all NEC cases require surgical intervention [39,40], which can lead to complications such as intestinal strictures and short bowel syndrome [41], and 61% of survivors exhibit significant neurodevelopmental delays [42].

A meta-analysis of randomized controlled trials (RCTs) demonstrated that both exclusive and partial human milk feeding significantly reduce the incidence of surgical NEC in preterm infants, particularly in those receiving a high proportion of breast milk [34].

Similarly, a multicenter randomized controlled trial involving extremely preterm infants found that those fed an exclusive preterm formula diet had prolonged dependence on parenteral nutrition and a higher incidence of surgical NEC compared to infants fed human milk [29].

Even mild cases of NEC are associated with impaired infant growth, increased antibiotic exposure, delayed achievement of full enteral feeding, higher healthcare costs, and prolonged NICU stays [27,32,33]. The bioactive components of breast milk have been shown to reduce intestinal inflammation, enhance stem cell proliferation, decrease enterocyte apoptosis, promote the development of a healthy microbiome, and mitigate oxidative stress, thereby addressing the multifactorial pathogenesis of NEC [5,43,44,45,46,47,48].

Recent meta-analyses [2,6,26,27] and individual studies [5,29,30,31,32,33] indicate that the use of human milk (maternal or donor) compared to preterm formula reduces the overall incidence of NEC by 50% to 75%, with a significant reduction in the risk of surgical NEC [13,22,34]. Clinical experience in NICUs strongly supports this approach [49,50].

The protective effect of human milk against NEC is dose-dependent, with greater benefits observed with exclusive or predominant use of human milk compared to preterm formula [2,51].

In addition to NEC, evidence suggests that the use of breast milk is associated with a reduced incidence of other complications of prematurity, including sepsis, BPD, and ROP [2,13,52,53,54,55,56].

The aim of this study was to compare infant outcomes in relation to the type of milk consumed, particularly in light of the implementation of a feeding protocol that included an exclusive human milk-based diet, while adjusting for confounding factors.

## 2. Materials and Methods

### 2.1. Study Design and Subjects

This is a single-center retrospective study. The cohort consisted of newborns admitted to the Neonatal Intensive Care Unit (NICU) at ‘Casa Sollievo della Sofferenza’ (San Giovanni Rotondo, Italy) between 2005 and 2019.

Inclusion criteria: Very low birth weight (VLBW) infants with a birth weight (BW) < 1500 g and gestational age (GA) < 32 weeks, in the absence of congenital anomalies or other unrelated conditions requiring transfer to other centers. Infants who died within the first week of life were also excluded from the analysis.

Data collection: For each subject, data regarding birth weight, gestational age, sex, type of delivery (vaginal or cesarean), single or multiple gestation, Apgar score, prenatal corticosteroid therapy, need for resuscitation, and duration of mechanical ventilation were collected. Data on patent ductus arteriosus (PDA) and the number of transfusions were also included. Data are expressed as mean values, median values, standard deviation, frequencies, and percentages.

Data sources: Data were extracted from the NICU database (NeoCare) and from clinical records available on the SISWEB system (National Health Information System), which provides access to the digitized medical records of discharged patients.

Feeding practices were classified into the following categories: infants fed exclusively with PF, infants receiving PF as a supplement, infants fed exclusively with MOM, infants fed predominantly with MOM, infants fed predominantly with DHM, and infants fed exclusively with DHM. For further analysis, these categories were grouped into three major groups: human milk, mixed milk, and formula milk.

Enteral feeding was advanced according to a standardized protocol. Fortifier was added to MOM and DHM when the enteral feeding volume reached 80 mL/kg/day. A feeding type was considered predominant when it exceeded 50% of the total intake. In cases of complementary feeding, the first type of milk reported was considered the predominant one.

Definitions: “Exclusive human milk” refers to feeding with MOM + DHM or DHM + MOM, “Any MOM” refers to feeding with MOM + preterm formula (PF) or PF + MOM, “Mixed milk” refers to feeding with MOM + PF.

The outcomes evaluated included NEC, ROP, BPD, and LOS. Data are expressed as frequencies and percentages. Each medical diagnosis was independently confirmed by multiple neonatologists.

LOS was defined as clinical signs and symptoms consistent with sepsis occurring more than 5 days after birth, in association with the isolation of a causative organism from a blood culture. BPD was diagnosed based on the need for oxygen at 36 weeks of postmenstrual age.

Data were analyzed both before and after the implementation of a feeding protocol that included an exclusive human milk-based diet.

Morbidities were considered multifactorial, with numerous covariates. Outcomes were assessed after adjusting for the type of care, which varied based on factors such as presence or absence of corticosteroid prophylaxis, GA, BW, Apgar score, type of resuscitation, type of respiratory support (defined as the duration in hours of invasive mechanical ventilation and non-invasive ventilation) PDA treatment, and number of transfusions.

The study was approved by the hospital’s Technical-Scientific Committee (TSC). The TSC determined that regional ethics committee review was not required, as data collection was conducted in full compliance with anonymization protocols and in accordance with TSC guidelines.

### 2.2. Statistical Analysis

Descriptive statistics were performed using Jamovi^®^ Statistical Software Version 2.3. Quantitative variables are expressed as means and standard deviations, while qualitative variables are presented as frequencies and percentages.

The incidence of morbidities was calculated across feeding categories, and frequencies were compared using Fisher’s exact test due to the small sample size. Differences between means were assessed using *t*-tests. To study the association between type of feeding and comorbidities, Spearman’s correlation was performed. Subsequently, the results were adjusted for confounding factors using partial correlation.

Spearman’s test was also used to study the association between ROP, BPD, and risk factors. To confirm the association between outcomes and feeding categories, odds ratios were calculated. Trends in BPD incidence and feeding practices were described using percentages. A *p*-value < 0.05 was considered statistically significant.

## 3. Results

### 3.1. Study Population

The characteristics of the study population are presented in Table 1 and Table 2. Based on gestational age (GA), the sample consisted of 57 neonates with a GA of ≤25 weeks, 223 neonates with a GA between 26 and 30 weeks, and 57 neonates with a GA > 30 weeks. Of these, 77 (22.8%) were delivered via spontaneous vaginal delivery.

Based on birth weight, 165 infants were classified as extremely low birth weight (ELBW) (≤1000 g). Among the VLBW infants, 195 (60.7%) received complete steroid prophylaxis, 50 (15.4%) received a single dose, and 76 (23.4%) received no steroid prophylaxis.

Regarding Apgar scores, 185 infants had a score of ≤5 at one minute, while 13 had a score of ≤5 at five minutes. Invasive resuscitation with endotracheal intubation (ET) was administered to 51.7% of infants, a rate that has decreased by half over the years, and 46.7% received resuscitation using a NeoPuff with a mask.

Hemodynamically significant patent ductus arteriosus (PDA) was observed in 56 (16.6%) VLBW infants. Pharmacological closure was achieved in all cases except for two infants, who required surgical intervention.

### 3.2. Feeding Practices

Among the 337 included infants, the distribution of infants based on feeding type was as follows; in total, 112 (33.3%) received exclusive preterm formula, 51 (15%) received MOM, 75 (22.1%) received DHM, 26 (7.6%) received a combination of MOM and PF, 39 (11.5%) received MOM + DHM, and 17 (5%) received DHM + MOM.

### 3.3. Incidence of Morbidities

The overall incidence of NEC was 4.2%. The incidence of ROP across all stages was 24.3%, with severe ROP observed in 16 cases (4.7%). The incidence of LOS was 5.6% and BPD across all stages was 25.5%, with moderate-to-severe BPD occurring in 22.2% of cases (75 infants).

### 3.4. Main Outcome

In the analysis of outcome incidence, the incidence of surgical NEC was 10% among infants fed exclusively with preterm formula, completely absent in those fed exclusively or partially with MOM, and 1.3% in those fed with DHM. These differences between feeding groups were statistically significant (*p* < 0.001) (Table 3, Table 4 and Table 5; Figure 1 and Figure 2).

Exposure to formula milk was associated with an increased risk of severe ROP, with rates ranging from 15% to 65%, compared to exclusive or any human milk feeding. The most significant comparison was between DHM and formula feeding, as no cases of ROP were observed among the 75 infants fed exclusively with DHM.

The rate of LOS was lower in infants fed with human milk, either MOM or DHM, with rates of 7.8% and 3.8%, respectively, compared to the 10% incidence observed in formula-fed infants.

The incidence of moderate-to-severe BPD varied across feeding subgroups. It was 17.5% in infants fed with MOM, 18.1% in those fed with formula, 21.3% in those fed with DHM, 12% in those fed with exclusive human milk, and 8.1% in those fed with any MOM.

The NNT (number needed to treat) value (Table 3), considering the frequency % of the outcome in relation to risk exposure, represented by feeding with PF, was found to be lower (between 10 and 20) for NEC, for LOS in the comparison between feeding with PF and any MOM/any HM, for severe ROP in the comparison between HMD and PF, and for severe BPD in the comparisons of PF vs. HM, Any MOM vs. PF, and Any HM vs. PF.

The associations between NEC and feeding categories—defined as human milk (MOM + DHM), preterm formula (PF), and mixed milk (MOM + PF, DHM + PF)—were statistically significant (Figure 1, Table 4). This includes the correlation adjusted for BW, GA, resuscitation, and Apgar scores at 1 and 5 min (Table 5), where statistical significance for LOS is also evident in the comparison between different categories of milk feeding.

In the correlation between outcomes and type of feeding (Table 6), the partial correlation analysis revealed an association for NEC and LOS when adjusted for covariates such as GA, BW, hours of synchronized intermittent positive pressure ventilation (SIPPV), Apgar scores at 1 and 5 min, and the presence of hemodynamically significant PDA requiring treatment.

Significance was also observed for other outcomes when the correlation was analyzed between severe forms of these conditions and the various types of feeding. Specifically, comparisons were made between MOM vs. PF, DHM vs. PF, Exclusive HM vs. PF, and Any MOM vs. DHM using odds ratio (OR) calculations (Table 7).

A significant correlation was also observed between ROP and risk factors, including corticosteroid prophylaxis, the duration of both invasive and non-invasive ventilation, and the number of transfusions (Table 8).

The average duration of invasive ventilation (in hours) was 218 for infants fed with MOM, 236 for those fed with DHM, and 338 for VLBW infants fed with PF (Table 9). Lower values were observed when feeding was exclusively human milk-based (Figure 2).

Regarding BPD, the incidence has increased over the years, reaching a peak of 46.6% in 2018 (Figure 3). A statistically significant correlation was found between BPD and variables such as GA, BW, and type of ventilation (Table 10).

### 3.5. Secondary Outcome

The other endpoint focused on the rates of feeding with MOM, which increased over the years, alongside a progressive decline in the use of DHM (Table 11; Figure 4 and Figure 5).

The rate of exclusive MOM feeding increased from an average of 10% in the years up to 2011 to 35% in 2017 and 45% in 2018–2019. The use of DHM declined from an average of 66.5% prior to 2014 to an average of 24% in the following years.

Complementary feeding increased from 21.7% in 2013 to 36.6% in 2019, with MOM being the predominant component.

## 4. Discussion

Although the limited data do not allow for a highly significant comparison for all outcomes, the trend of the impact of feeding type in favor of human milk emerges from this study with some particularly significant correlations.

One of the key roles of neonatal units is to support the growth and development of preterm infants during the critical period between birth and the equivalent of full-term age. Nutritional support is a crucial part of this process, and human milk significantly contributes to this goal. Not only is it the most suitable option, as it is produced by the infant’s own mother during that specific gestational phase, but it also has direct and indirect effects on growth and development, mitigating the risk of complications associated with preterm birth. Human milk is the first choice for preterm infants; however, it is not always available from the first hours or throughout the initial weeks of life in the NICU.

There are limited data monitoring these aspects [57,58], which would be highly valuable. One of the few multicenter studies, referencing an investigation conducted across 11 European countries, revealed that, on average, only 58% of very low birth weight infants received human milk at discharge, and this was not exclusive [59]. The rates varied between 36% and 80%. These findings are consistent with data from China (58%) [60] and the United States [61,62,63], where rates range from 30% to 70–75%.

Our data also align closely with these figures. However, this means that approximately 50% of VLBW infants, who are most in need, do not receive mother’s own milk. In such cases, donor human milk should be the second choice, as, despite undergoing treatments to ensure microbiological safety, it provides VLBW infants with essential nutritional and bioactive components to support extrauterine development [28]. Additionally, it minimizes exposure to cow-milk-based formulas, which have been associated with an increased risk of NEC.

Although the practice of human milk banking is expanding globally, with over 800 milk banks currently operating across 66 countries [64], and despite official endorsements from international associations recommending the use of DHM as an alternative to MOM [65,66], a full consensus has yet to be achieved. This is due to the lack of conclusive evidence demonstrating the superiority of donor milk over preterm formulas in addressing all complications associated with prematurity, as well as the limited availability of RCTs. Such trials are challenging to conduct, primarily due to ethical concerns regarding the deprivation of human milk in preterm infants solely for research purposes. In reality, research in this field has been ongoing for over 50 years.

From the first early case–control study [67], prompted by the observation that ROP appeared to occur less frequently in infants fed human milk in a context where the primary rationale for promoting MOM in NICUs was solely to encourage maternal involvement in the care of very preterm infants, much has changed. A turning point came with the study by Lucas et al. [68] in the mid-1980s, which provided the largest dataset at the time on the effects of MOM, DHM, and formula feeding in preterm infants [69], significantly influencing best practices in neonatal intensive care regarding this aspect of care [51,70].

Over the past two decades, interest has increasingly centered on practices that influence the risk of NEC, LOS, BPD, and ROP. These are complications linked to prematurity that are potentially preventable during NICU hospitalization through a theoretically more feasible method such as MOM feeding, as evidenced by several studies. Furthermore, these morbidities are strongly associated with an elevated risk of neurodevelopmental disorders and chronic health conditions in childhood, as highlighted by higher rates of rehospitalization and the need for specialized educational support in the VLBW population [71].

In 2019, in the meta-analysis by Quigley et al. [26] the use of formula was associated with an increased risk of developing NEC compared to donor human milk.

Despite limitations due to small subgroup sizes, which may reduce statistical power and pose challenges in interpretation, all results including those related to the NNT suggest a protective effect of human milk against NEC in all feeding regimens that include HM. Additionally, the presence of MOM significantly influences the risk of LOS, though donor milk also substantially reduces this risk. For ROP, donor milk showed the most pronounced protective effect, while the comparison with MOM did not reach significance, likely due to the limited sample size. In cases of severe BPD, the use of HM, compared to PF, was associated with a low NNT, highlighting its beneficial impact.

### 4.1. NEC

Mixed feeding, which combines HM with PF, is a widely adopted practice for pre-term infants when an exclusive human milk diet is not feasible due to insufficient supply.

Recent meta-analyses [2,72,73], as well as multiple randomized controlled trials (RCTs) and observational studies, have demonstrated significant reductions in the incidence of NEC among very preterm infants fed a human-milk-based diet compared to those receiving a formula-based diet. These studies report reduction rates in NEC ranging from 50% to 100%. In our study it was about 100%.

A study [74] across seven U.S. centers revealed that NICUs implementing exclusive human-milk-diet programs observed a reduction or variation in overall NEC rates (medical and surgical), with surgical NEC rates declining by 66% to 100%.

Our prior research [49] reported a 100% reduction in NEC incidence following the introduction of donor milk and discontinuation of formula—a finding confirmed by subsequent evidence.

Our current results highlight the profound impact of use of exclusive HM on NEC. The adjusted, statistically significant data, despite limited sample size, clearly show a protective effect against surgical NEC. This strongly supports promoting practices to enhance milk production in mothers of VLBW infants and using donor milk when maternal milk is unavailable. Additionally, infants transferred for surgical NEC exhibited high mortality rates or significant long-term complications.

Over the five-year period during which VLBW infants were fed formula as an alternative to maternal milk, the incidence of surgical NEC was 6.7%, consistent with data from the Vermont Oxford Network (VON). However, the incidence of NEC varies across studies, ranging from 1.3% to 12.9% in large cohort studies by Yee et al. [75] and from 3% to 12% in those by Fitzgibbons et al. [76]. Other observational studies [77,78] report an incidence ranging from 16.7% [4,5] to 20.7%, independent of risk factors and care-related variables.

It is important to emphasize that all cases of NEC occurred in VLBW infants exclusively fed formula. The sole exception was an infant who had initiated donor milk feeding: an extremely high-risk preterm born at 24 weeks’ gestational age with a birth weight of 700 g. This infant lacked antenatal corticosteroid prophylaxis, had early-onset Klebsiella sepsis, maternal chorioamnionitis, and grade IV-IVH (intraventricular hemorrhage) diagnosed at birth. Intestinal distress developed within the first week of life despite minimal enteral feeding. Additionally, two VLBW infants developed NEC without ever initiating enteral feeding, further supporting the multifactorial etiology of the disease.

In light of these findings, we conclude that the exclusive use of human milk during the early weeks of life is an effective strategy to significantly reduce the risk of NEC.

When provided as an exclusive diet or in combination with maternal milk, pasteurized donor milk is also protective against NEC. However, there is still a lack of definitive evidence demonstrating the same unequivocal health benefits as maternal milk, such as reductions in LOS, ROP, and BPD [26].

### 4.2. ROP

In our case series, 58% of severe ROP cases involved VLBW infants fed with formula and the introduction of donated human milk resulted in a halving of the incidence of severe ROP requiring ophthalmological treatment. Specifically, the partial correlation between outcomes and feeding type, adjusted for covariates, was not significant for ROP. However, significance was observed when analyzing the correlation between severe forms of ROP and various feeding types (OR < 0.001).

It is possible—and this is a personal interpretation—that the limited sample sizes intended for comparison may have constrained the results and their interpretation.

As might be expected, a significant correlation was also observed between ROP and risk factors such as corticosteroid prophylaxis, the duration of both invasive and non-invasive ventilation, and the number of transfusions. These findings remain valuable in guiding care practices in the NICU.

In previous studies, among VLBW infants, human milk feeding was associated with a reduced incidence of ROP compared to exclusive formula feeding, after adjusting for confounding variables [79].

In 2013, Manzoni et al. [9] tested the hypothesis that exclusive feeding with fresh maternal milk could prevent ROP in VLBW infants compared to formula feeding. After adjusting for potential confounding factors, it was found that exclusive human milk feeding from birth could prevent ROP of any stage in VLBW infants in the NICU.

Both in observational studies and meta-analyses, albeit based on the currently limited evidence, a protective role of any amount of human milk in the prevention of ROP at any stage and of severe ROP in VLBW infants and extremely low birth weight infants (ELBWIs) is recognized [2,5].

A 2015 meta-analysis [8] included five observational studies that compared the incidence of any stage and severe ROP in infants fed human milk versus formula and revealed that exclusive or predominant human milk feeding protects against any ROP and severe ROP.

Three other meta-analyses [9,80,81,82,83] found a protective effect of human milk on the development of ROP.

Therefore, although some conflicting results have been reported [8,84], the prevailing literature includes the use of human milk among the protective factors against ROP.

### 4.3. LOS

Our data confirm the ability of human breast milk to mitigate most morbidity in the NICU, including the risk of LOS. A clear statistical significance was observed in the analyses related to LOS. In fact, in the correlation between outcomes and feeding type (Table 6), partial correlation analysis revealed an association for LOS when adjusted for covariates. This significance was also confirmed in the partial correlation between outcomes and feeding type, as well as feeding categories and was further supported by Spearman’s correlation analysis (Table 3 and Table 4).

The incidence of any infection, as well as sepsis/meningitis, is significantly reduced in VLBW infants fed human milk compared to those exclusively fed formula [79]. This specific effect has been contested by the de Silva group [85], who concluded in their systematic review that the benefit of human milk in preventing infections in preterm infants (VLBW) is not supported by existing studies.

This assertion has been challenged by other authors [86], who highlight that the effect is dose dependent. VLBW infants receiving more than 50 mL/kg/day of human milk during the first four weeks of life exhibit a reduced rate of sepsis, confirming that feeding with human milk is beneficial for these infants.

This claim is supported by Patel [7], who aimed to evaluate the costs associated with morbidity in the neonatal unit and demonstrated a dose–response relationship between exposure to human milk during days 1–28 and a reduction in the likelihood of sepsis, after controlling for propensity score. For every 10 mL/kg/day increase in human milk intake, the odds of sepsis decreased by 19%.

This superiority has not been demonstrated in a randomized clinical trial [87] specifically comparing pasteurized donated milk to preterm formula as supplementary feeding during the first 10 days of life in very low birth weight neonates; however, the short duration of the study limited its effectiveness.

In Miller’s meta-analysis [2] of studies published after 1990 on morbidity rates in preterm infants with various types of feeding, alongside evidence of a clear protective effect of high doses of human milk against NEC, a possible reduction in the risk of developing LOS and severe ROP was also reported in infants exclusively fed human milk.

Some authors [88], while observing a positive effect on LOS, agree on the need for a greater number of RCTs with larger sample sizes.

It appears more widely demonstrated and accepted that the incidence of this complication decreases with higher doses of human milk [22] and with the early initiation of human milk feeding [89].

### 4.4. BPD

In studies [90] where the outcomes of extremely low birth weight infants were evaluated following the introduction of donor milk to supplement mother’s milk, an association was also found between exposure to formula in preterm infants and the need for respiratory support, which is closely linked to the development of BPD.

Our analysis shows no significant associations between BPD and feeding categories or types, even after adjusting for relevant factors. However, the odds ratio (OR) was significant (*p* < 0.001) for the association between MOM vs. DF, DHM vs. PF, Exclusive HM vs. PF, and Any MOM vs. DHM in relation to the incidence of severe BPD.

In addition, two key observations emerge from our case series. First, there is a reported increase in the overall incidence of this prematurity-related complication, consistent with the literature [91,92]. Second, the type of feeding influences the duration of ventilation (Table 9), with shorter mean periods observed in exclusively human milk-based feeding (Figure 3).

Bronchopulmonary dysplasia is a chronic lung disease of preterm infants and is characterized by abnormal lung development caused by factors such as mechanical ventilation, oxygen therapy, or poor nutrition [91,92]. Various definitions for BPD have been proposed [93,94]. The most recent of these definitions no longer includes the criterion that oxygen therapy must be required at 28 days of life and instead focuses solely on the increased need for oxygen at 36 weeks of postmenstrual age [95], which we considered in the study.

Advances in many aspects of care, such as avoiding invasive ventilation and fluid overload [96], the use of caffeine [97], and late systemic corticosteroids [98] in infants who are unable to wean off mechanical ventilation, are evidence-based strategies to reduce the potential development of BPD [99]. However, despite documented reductions in most neonatal morbidities, the incidence of BPD remains unchanged or is increasing [100,101]. Our current data also confirm this (Figure 4).

In fact, advances in the care of preterm infants and the increased survival of the most immature infants have led to a rise in the incidence of BPD, which has become the most common morbidity following preterm birth.

The incidence of this disease increases with greater prematurity, with studies reporting grade 2–3 BPD in up to 55% of infants born at <28 weeks’ gestation [102], consistent with the 50% reported by the Vermont Oxford Network for infants born at <30 weeks [103,104]. Data from China between 2010 and 2019 indicate that the incidence of BPD among extremely premature newborns exceeded 74% [105].

In our data, this association was confirmed, as was the one with BW and the duration of various types of ventilation.

Some studies have demonstrated that VLBW infants who are exclusively formula-fed exhibit a significantly higher risk of developing BPD [106]. Conversely, a high proportion of HM in the diet of these infants has been consistently associated with a reduced risk of BPD, as supported by multiple studies [12,104,105,106,107].

Given the absence of effective treatments, prevention strategies are critical for pre-term infants at risk of BPD. Breastfeeding with MOM has proven effective in preventing severe BPD [4,5,6]. This protective effect is partly attributed to breast milk components reducing oxidative stress, which plays a key role in the inflammation and alveolar destruction linked to dysplasia [6,7,8].

Therefore, there is also strong support for the recommendation of MOM as the primary source of nutrition for this type of morbidity of prematurity, particularly for VLBW infants [9,10].

The literature is less conclusive regarding the comparison between MOM and DHM in terms of their effect on BPD [11]. Nevertheless, based on the available data, the prevailing consensus supports the use of exclusively human milk as an alternative when maternal milk is unavailable, demonstrating a similar protective effect in the prevention of severe BPD [13,106,107].

For this complication as well, HM volume plays a significant role. Higher HM intake is associated with a lower risk of BPD compared to lower HM volumes or formula feeding [104,105,106,107]. Analyzing effects based on the quantities of maternal and donor HM could further clarify these correlations. Additionally, HM improves feeding tolerance, enabling faster enteral advancement and reduced duration of parenteral nutrition, both of which significantly impact the development and progression of BPD [31,108].

### 4.5. MOM Feeding

Regarding the second endpoint, which focused on the rates of feeding with MOM, the findings confirm what has already been documented in a previous study on the topic. The rates increased from values consistently below 20% in the years up to 2011 to over 40% in the years 2017–2019. Concurrently, there was a reduction in the exclusive use of DHM, which decreased from an average of over 70% in the years 2011–2013 to 24% in the years 2015–2019. On the other hand, donor milk should be considered a nutritional “bridge” until full feeding with MOM can be achieved.

These data suggest that a milk bank may promote MOM feeding in VLBW infants, a critical factor given MOM’s well-documented benefits to mortality and health outcomes in this population.

For this outcome, with the adoption of advanced donor milk processing methods in milk banks—better preserving bioactive factors akin to fresh MOM—HM has been established as the standard for preterm infant nutrition by health authorities [24,65], pending further definitive evidence on donor milk’s protective effects on preterm mortality and morbidity.

## 5. Conclusions

The data confirm the use of human milk as a crucial element in the prevention of NEC and LOS, its protective effect against severe ROP, and its role in promoting feeding with MOM, with rates increasing significantly when DHM is available. Ensuring access to human milk for all VLBW infants should become the standard of care in all NICUs.

## 6. Strengths and Limitations

The strengths of the study include the analysis being adjusted for confounding factors and the ability to compare outcomes before and after the introduction of donor human milk in the NICU. The limitations include its single-center design and the low overall incidence of these morbidities, which restricts the statistical analyses that can be employed to model the risk of morbidity as a function of feeding type.

## Figures and Tables

**Figure 1 nutrients-17-01138-f001:**
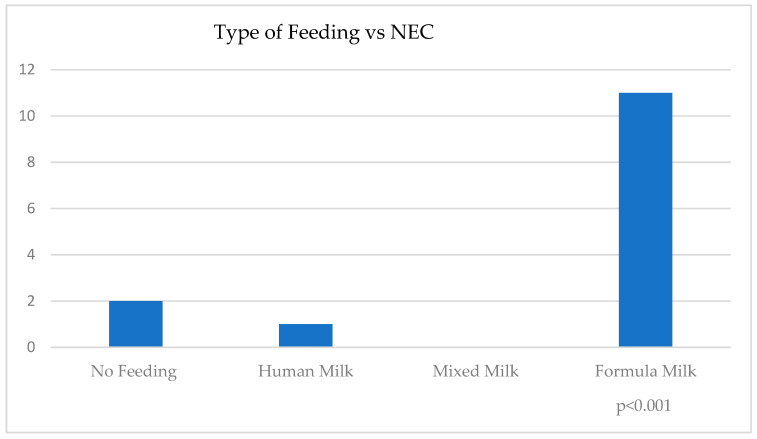
The associations between NEC and feeding categories, expressed as Fisher’s exact test *p* < 0.001. NEC—necrotizing enterocolitis; Human Milk—MOM + DHM; PF—preterm formula; Mixed Milk—MOM + PF, DHM + PF.

**Figure 2 nutrients-17-01138-f002:**
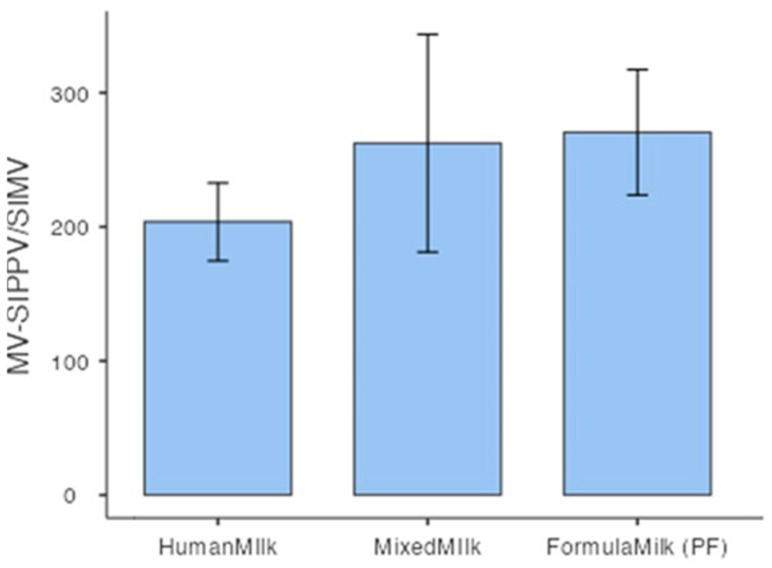
MV duration and feeding class. Graphical representation of the duration of invasive mechanical ventilation in hours distributed by type of feeding. *X*-axis: type of feeding. *Y*-axis: duration of mechanical ventilation in hours. SIPPV/SIMV—synchronized intermittent positive pressure ventilation/synchronized intermittent mandatory ventilation; Human Milk—MOM + DHM; Mixed Milk—MOM + PF, DHM + PF; MOM—mother’s own milk; PF—preterm formula; DHM—donor human milk; MV—mechanical ventilation.

**Figure 3 nutrients-17-01138-f003:**
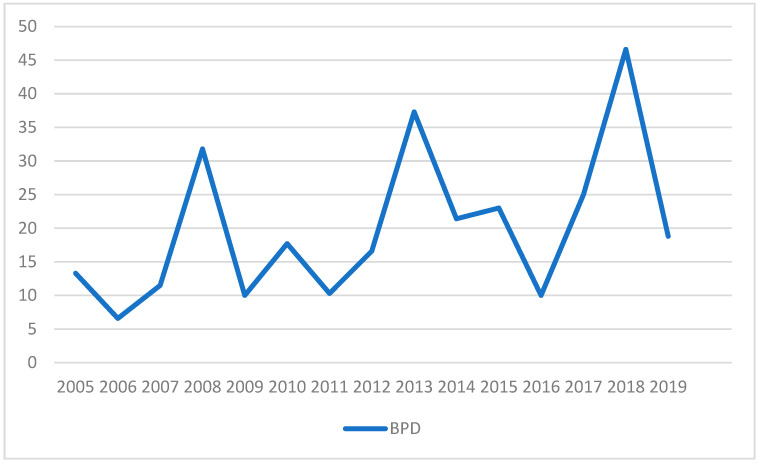
Trend of BPD incidence. The table illustrates the temporal trends in the incidence of bronchopulmonary dysplasia (BPD) over the years 2005 to 2019. The data provide an overview of how the prevalence or incidence of BPD has changed over time.

**Figure 4 nutrients-17-01138-f004:**
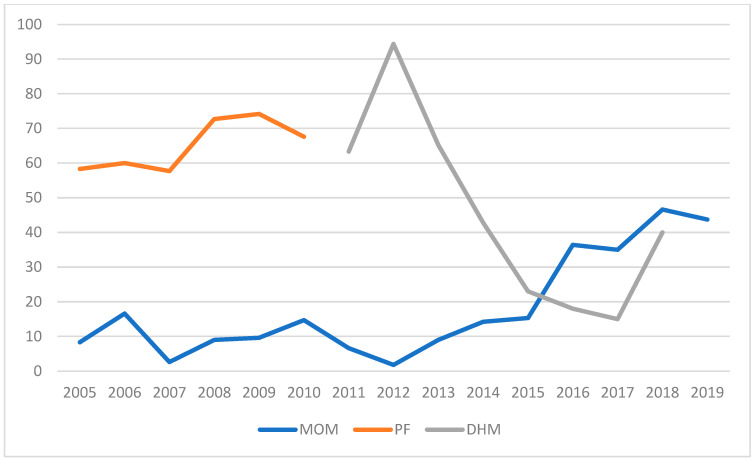
Feeding trend. Trend of average percentages of feeding with MOM, PF, and DHM from 2005 to 2019. MOM—mother’s own milk; PF—preterm formula; DHM—donor human milk.

**Figure 5 nutrients-17-01138-f005:**
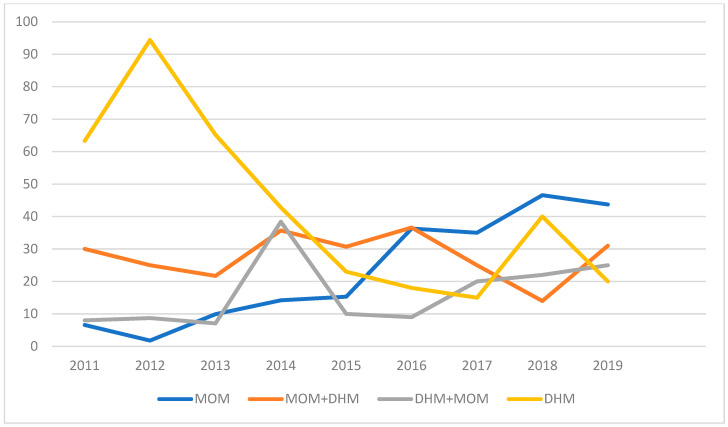
Trend of human-milk use. The table illustrates the temporal trends in the use of human milk over a specified period. The data provide an overview of how the prevalence or proportion of human milk feeding has changed over time, reflecting potential shifts in clinical practices, parental preferences, or institutional policies. MOM—mother’s own milk; DHM—donor human milk.

**Table 1 nutrients-17-01138-t001:** Sample description. Summary of the sample characteristics.

	Variable	
N: 337	Gestazional age, weeks {M(SD)}	27.9 (2.51)
	Birth weight, g {Median (SD)}	1040 (266)
	Female {n (%)}	172 (51)
	Multiple gestation {n (%)}	118 (35)
	Type of delivery {n (%)}	
	- Vaginal delivery	77 (22.8)
	- C-section	123 (36.5)
	- Emergency C-section	137 (40.7)
	Neonatal resuscitation {n (%)}	313 (92.9)
	- ET	163 (51.7)
	- Neopuff	147 (46.7)
	- Other	3 (0.9)
	Apgar 1′ {M(SD)}	5 (1)
	Apgar 5′ {M(SD)}	7 (1)
	Feeding {n (%)}	
	- Esclusive Human Milk	182 (54.2)
	- Mixed Milk	40 (11.9)
	- Preterm Formula	112 (33.3)
	- No feeding	3 (0.6)
	Mechanical ventilation hours {M(SD)}	
	- SIPPV/SIMV	232 (360)
	- nIMV	251 (346)
	- nCPAP	299 (314)
	Any neonatal complication {n (%)}	
	- NEC	14 (4.2)
	- BPD	86 (25.5)
	- ROP	82 (24.3)
	- LOS	19 (5.6)

NEC—necrotizing enterocolitis; ROP—retinopathy of prematurity; LOS—late-onset sepsis; BPD—bronchopulmonary dysplasia; IMV: intermittent mandatory ventilation; SIPPV/SIMV—synchronized intermittent positive pressure ventilation/synchronized intermittent mandatory ventilation; CPAP—continuous positive airway pressure.

**Table 2 nutrients-17-01138-t002:** Severe outcomes incidence. Incidence of severe outcomes across different feeding categories.

Subgroup	Sample n	Severe BPD n (%)	Severe ROP n (%)	Surgical NEC n (%)	LOS n (%)
MOM	51	9 (17.6)	2 (3.9)	0 (0)	4 (7.8)
PF	110	20 (18.2)	5 (4.5)	11 (10)	11 (10)
DHM	75	16 (21.3%)	0 (0)	1 (1.3%)	3 (4%)
HM escl.	182	22 (12)	5 (2.74)	1 (0.5)	3 (4)
Any MOM	37	3 (8.1)	2 (5.4)	0 (0)	1 (2.7)
Any HM	40	5 (12.5)	2 (5)	0 (0)	2 (5)
No Feeding	3			2	

The table displays the incidence rates of severe clinical outcomes stratified by different feeding categories. HM escl. (MOM/DHM/MOM + DHM/DHM + MOM), Any MOM (MOM + PF/PF + MOM), Any HM (MOM + PF/PF + MOM/DHM + PF/PF + DHM). MOM—mother’s own milk; PF—preterm formula; HM—human milk; DHM—donor human milk; NEC—necrotizing enterocolitis; ROP—retinopathy of prematurity; LOS—late-onset sepsis; BPD—bronchopulmonary dysplasia.

**Table 3 nutrients-17-01138-t003:** NNT (number needed to treat).

NNT Value				
	NEC	LOS	Severe ROP	Severe BPD
MOM vs. PF	10	45	166	166
DHM vs. PF	11	26	22	/
HM escl. vs. PF	10	26	55	16
Any MOM vs. PF	10	13	/	9
Any HM vs. PF	10	20	/	17

The NNT value is expressed as the reciprocal of the absolute risk reduction, represented as a percentage. The risk group being compared consists of VLBWs (very low birth weight infants) exposed to PF (preterm formula). HM escl. (MOM/DHM/MOM + DHM/DHM + MOM), Any MOM (MOM + PF/PF + MOM), Any HM (MOM + PF/PF + MOM/DHM + PF/PF + DHM). MOM—mother’s own milk; HM—human milk; DHM—donor human milk; NEC—necrotizing enterocolitis; ROP—retinopathy of prematurity; LOS—late-onset sepsis; BPD—bronchopulmonary dysplasia.

**Table 4 nutrients-17-01138-t004:** Spearman’s correlation between NEC, LOS, mechanical ventilation, and type of feeding and feeding classis.

Spearman’s Rho		Type of Feeding (No Feeding/DHM + MOM/PF + MOM/ MOM/PF/DHM/MOM + PF/MOM + DHM/DHM + PF)	Feeding Class Human Milk/Mixed Milk/Formula Milk
NEC	Correlation coefficient	0.180 ***	0.141 **
	*p* value	<0.001	0.010
LOS	Correlation coefficient		−0.110 *
	*p* value	Ns	0.044
MV—SIPPV/SIMV	Correlation coefficient		0.173 **
	*p* value	Ns	0.009

The table presents Spearman’s rank correlation coefficients between necrotizing enterocolitis (NEC), late-onset sepsis (LOS), duration of mechanical ventilation, and the type of feeding or feeding categories. Human Milk—MOM + DHM; Mixed Milk—MOM + PF, DHM + PF; MOM—mother’s own milk; PF—preterm formula; HM—human milk; DHM—donor human milk; NEC—necrotizing enterocolitis; LOS—late-onset sepsis; MV—mechanical ventilation, SIPPV/SIMV—synchronized intermittent positive pressure ventilation/synchronized intermittent mandatory ventilation. (hours); * *p* < 0.05; ** *p* < 0.01; *** *p* < 0.001.

**Table 5 nutrients-17-01138-t005:** Partial correlation between outcomes and type of feeding and feeding classis.

Spearman’s Rho		Type of Feeding (No Feeding/DHM + MOM/PF + MOM/ MOM/PF/DHM/MOM + PF/MOM + DHM/DHM + PF)	Feeding Classis Human Milk/Mixed Milk/Formula Milk
NEC	Correlation coefficient	0.186 ***	0.149 **
	*p* value	<0.001	0.007
LOS	Correlation coefficient		−0.118 *
	*p* value	Ns	0.032

The table displays partial correlation coefficients between clinical outcomes and the type of feeding or feeding categories, adjusted for potential confounding variables. The analysis isolates the specific relationships between feeding practices and outcomes by controlling for other influencing factors, providing a clearer understanding of their independent associations. Adjusted for birth weight, gestational age, resuscitation, Apgar score 1′ and 5′. * *p* < 0.05; ** *p* < 0.01; *** *p* < 0.001. Human Milk—MOM + DHM; Mixed Milk—MOM + PF, DHM + PF; MOM—mother’s own milk; PF—preterm formula; DHM—donor human milk; NEC—necrotizing enterocolitis; LOS—late-onset sepsis.

**Table 6 nutrients-17-01138-t006:** Partial correlation between outcomes and type of feeding.

MOM/PF/DHM/HM escl./Any MOM	NEC	ROP	LOS	BPD
Correlation coefficient	<0.001	0.9551	<0.001	0.1691

The table presents the partial correlation coefficients, adjusted for potential confounding variables, between clinical outcomes and the type of feeding. The analysis accounts for covariates to isolate the specific relationship between feeding modalities and outcomes. Adjusted for birth weight, gestational age, resuscitation, Apgar score 1′ and 5′, duration of invasive ventilation, PDB treatment, *p* < 0.001. MOM—mother’s own milk; PF—preterm formula; HM—human milk; DHM—donor human milk; NEC—necrotizing enterocolitis; ROP—retinopathy of prematurity; LOS—late-onset sepsis; BPD—bronchopulmonary dysplasia.

**Table 7 nutrients-17-01138-t007:** Odds ratio between milk categories.

	MOM vs. PF	DHM vs. PF	HM Exclusive vs. PF	ANY MOM vs. DHM
Chirurgical NEC	*p* < 0.001	*p* < 0.001	*p* < 0.001	*p* < 0.001
Severe ROP	*p* < 0.001	*p* < 0.001	*p* < 0.001	*p* < 0.001
LOS	*p* < 0.001			*p* < 0.001
Severe BPD	*p* < 0.001	*p* < 0.001	*p* < 0.001	*p* < 0.001

The table displays the odds ratios (OR) for the association between different types of milk feeding and the incidence of clinical outcomes. MOM—mother’s own milk; PF—preterm formula; HM—human milk; DHM—donor human milk; HM escl. (MOM/DHM/MOM + DHM/DHM + MOM), ANY MOM (MOM + PF/PF + MOM); NEC—necrotizing enterocolitis; ROP—retinopathy of prematurity; LOS—late-onset sepsis; BPD—bronchopulmonary dysplasia.

**Table 8 nutrients-17-01138-t008:** ROP and risk factors.

	Spearman’s Rho	MV-SIPPV/SIMV	n-IMV	n-CPAP	N Trasfusions	Prenatal Steroids
ROP	Correlation coefficient	−0.242 ***	−0.304 **	−0.379 ***	−0.326 ***	−0.141 *
	*p*-value	<0.001	0.001	<0.001	<0.001	0.010

The table presents the correlation coefficients (e.g., Spearman’s Rho) between retinopathy of prematurity and key clinical variables, including the type and duration of ventilation, the number of transfusions received, and exposure to prenatal steroids. Adjusted for corticosteroid prophylaxis, the duration of both invasive and non-invasive ventilation, and the number of transfusions. ROP—retinopathy of prematurity; MV—mechanical ventilation; IMV—intermittent positive pressure ventilation; SIPPV/SIMV—synchronized intermittent positive pressure ventilation/synchronized intermittent mandatory ventilation; CPAP—continuous positive airway pressure. * *p* < 0.05; ** *p* < 0.01; *** *p* < 0.001.

**Table 9 nutrients-17-01138-t009:** MV duration and the type of milk.

	MV Duration Hours Media (Min–Max)
HMB	236 (1–1440)
PF	338 (2–1440)
MOM + HMB	158 (2–1344)
MOM	218 (10–1464)
HM exclusive	207 (12–1440)
MOM + BF	233 (3–1320)

The table summarizes the duration of invasive mechanical ventilation, measured in hours, stratified by different feeding categories. MV—mechanical ventilation; MOM—mother’s own milk; PF—preterm formula; HMB—human milk bank; HM—human milk; DHM—donor human milk; HM escl.—(MOM/DHM/MOM + DHM/DHM + MOM).

**Table 10 nutrients-17-01138-t010:** Association between BPD and clinical variables.

	Spearman’s Rho	GA	BW	SIPPV/SIMV	n-IMV	n-CPAP
BPD	Correlation coefficient	0.487 ***	0.460 ***	−0.443 ***	−0.437 ***	−0.455 ***
	*p*-value	<0.001	<0.001	<0.001	<0.001	<0.001

The table presents Spearman’s rank correlation coefficients (ρ) between bronchopulmonary dysplasia (BPD) and gestational age (GA), birth weight (BW), and type of ventilation. BPD—bronchopulmonary dysplasia; GA—gestational age; BW—birth weight; IMV—intermittent positive pressure ventilation; SIPPV/SIMV—synchronized intermittent positive pressure ventilation/synchronized intermittent mandatory ventilation; cPAP—continuous positive airway pressure. *** *p* value < 0.001.

**Table 11 nutrients-17-01138-t011:** Feeding trend. Trend of average percentages of feeding with MOM, PF, and DHM from 2005 to 2019.

	MOM%	PF%	DHM%
2005	8.3	58.3	
2006	16.6	60	
2007	2.6	57.7	
2008	9	72.7	
2009	9.7	74.2	
2010	14.7	67.6	
2011	6.6		63.3
2012	1.8		94.4
2013	9		65.2
2014	14.2		42.8
2015	15.3		23
2016	36.3		18
2017	35		15
2018	46.6		40
2019	43.7		25

MOM—mother’s own milk; PF—preterm formula; DHM—donor human milk.

## Data Availability

The study data are available upon reasonable request from the corresponding author. The data are not publicly available due to privacy.

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
