# Peer review of "Impact of Enteral Nutrition on Clinical Outcomes in Very Low Birth Weight Infants in the NICU: A Single-Center Retrospective Cohort Study"

_nutrients, 2025, doi:10.3390/nu17071138_

Round 1
Reviewer 1 Report
Comments and Suggestions for Authors
An interesting and fascinating article worth publishing after referring to my comments:
Review of Manuscript: "Impact of Enteral Nutrition on Clinical Outcomes in Very Low Birth Weight Infants in the NICU"
Strengths of the Manuscript:
Relevant Topic & Clinical Importance: The study addresses an important issue in neonatology—feeding strategies for very low birth weight (VLBW) infants—which has significant implications for clinical practice.
Comprehensive Literature Review: The introduction and discussion provide a thorough overview of the current literature, highlighting the importance of human milk in neonatal care.
Statistical Analyses & Adjustments: The manuscript includes robust statistical analyses, adjusting for potential confounders, which strengthens the credibility of the findings.
Clear Findings: The results effectively demonstrate the protective effects of human milk in preventing necrotizing enterocolitis (NEC), late-onset sepsis (LOS), retinopathy of prematurity (ROP), and bronchopulmonary dysplasia (BPD).
Areas for Improvement:
1. Clarity & Structure:
Abstract: The abstract is well-structured but can be slightly more concise. Consider summarizing key numerical results in a clearer manner to improve readability.
Introduction: While informative, some parts of the introduction are redundant. The emphasis on the benefits of human milk is repeated multiple times—consider streamlining these sections to avoid redundancy.
Discussion: While comprehensive, some sections are lengthy and could be better structured into subsections focusing on each neonatal complication (NEC, LOS, ROP, BPD) with more concise interpretations.
2. Methodological Considerations:
Study Design Description: The study design could be more clearly described. The inclusion and exclusion criteria should be explicitly stated in a structured format.
Statistical Methods: The statistical section is thorough but could benefit from a brief explanation of why specific tests (e.g., Fisher’s exact test, Spearman’s correlation) were chosen for different variables.
Confounding Factors: While the study adjusts for confounders, it would be useful to elaborate on any potential unmeasured confounders that might have influenced the findings.
3. Data Presentation & Tables:
Tables & Figures Formatting: Some tables contain dense text, making them difficult to interpret at first glance. Consider using more spacing, highlighting key results, and ensuring consistent formatting across all tables.
Figure Legends: Some figures, such as Figures 1 and 2, should have clearer legends explaining what each comparison represents.
Statistical Significance Indicators: Ensure all p-values and confidence intervals are consistently reported across tables and figures.
4. Language & Grammar:
Grammar & Sentence Structure: Some sentences are overly complex. Shortening them would improve readability. Consider revising passive voice constructions for a more direct approach.
Terminology Consistency: Throughout the paper, ensure that terms such as “Mother’s Own Milk (MOM)” and “Donor Human Milk (DHM)” are used consistently.
5. Ethical Considerations:
Ethical Approval Statement: While it is mentioned that the study did not require ethical approval, it might be useful to clarify why it was waived and whether any ethical guidelines were followed in retrospective data collection.
Informed Consent: If patient data was used, a brief note on data anonymization and confidentiality would strengthen the ethical clarity.
Recommendations for Corrections:
Refine the abstract to make it more concise and impactful.
Streamline the introduction to avoid redundant information.
Clarify the methodology, particularly how outcomes were measured and how feeding categories were defined.
Improve table readability by reformatting them for clarity.
Ensure consistency in terminology throughout the paper.
Revise grammar and sentence structure for improved readability.
Enhance the ethical considerations section to explicitly mention ethical guidelines followed.
Overall Evaluation:
This is a well-conducted study with meaningful contributions to neonatal care. Addressing the above points will improve the clarity, coherence, and overall impact of the manuscript.
An interesting and fascinating article worth publishing after referring to my comments:
Review of Manuscript: "Impact of Enteral Nutrition on Clinical Outcomes in Very Low Birth Weight Infants in the NICU"
Strengths of the Manuscript:
Relevant Topic & Clinical Importance: The study addresses an important issue in neonatology—feeding strategies for very low birth weight (VLBW) infants—which has significant implications for clinical practice.
Comprehensive Literature Review: The introduction and discussion provide a thorough overview of the current literature, highlighting the importance of human milk in neonatal care.
Statistical Analyses & Adjustments: The manuscript includes robust statistical analyses, adjusting for potential confounders, which strengthens the credibility of the findings.
Clear Findings: The results effectively demonstrate the protective effects of human milk in preventing necrotizing enterocolitis (NEC), late-onset sepsis (LOS), retinopathy of prematurity (ROP), and bronchopulmonary dysplasia (BPD).
Areas for Improvement:
1. Clarity & Structure:
Abstract: The abstract is well-structured but can be slightly more concise. Consider summarizing key numerical results in a clearer manner to improve readability.
Introduction: While informative, some parts of the introduction are redundant. The emphasis on the benefits of human milk is repeated multiple times—consider streamlining these sections to avoid redundancy.
Discussion: While comprehensive, some sections are lengthy and could be better structured into subsections focusing on each neonatal complication (NEC, LOS, ROP, BPD) with more concise interpretations.
2. Methodological Considerations:
Study Design Description: The study design could be more clearly described. The inclusion and exclusion criteria should be explicitly stated in a structured format.
Statistical Methods: The statistical section is thorough but could benefit from a brief explanation of why specific tests (e.g., Fisher’s exact test, Spearman’s correlation) were chosen for different variables.
Confounding Factors: While the study adjusts for confounders, it would be useful to elaborate on any potential unmeasured confounders that might have influenced the findings.
3. Data Presentation & Tables:
Tables & Figures Formatting: Some tables contain dense text, making them difficult to interpret at first glance. Consider using more spacing, highlighting key results, and ensuring consistent formatting across all tables.
Figure Legends: Some figures, such as Figures 1 and 2, should have clearer legends explaining what each comparison represents.
Statistical Significance Indicators: Ensure all p-values and confidence intervals are consistently reported across tables and figures.
4. Language & Grammar:
Grammar & Sentence Structure: Some sentences are overly complex. Shortening them would improve readability. Consider revising passive voice constructions for a more direct approach.
Terminology Consistency: Throughout the paper, ensure that terms such as “Mother’s Own Milk (MOM)” and “Donor Human Milk (DHM)” are used consistently.
5. Ethical Considerations:
Ethical Approval Statement: While it is mentioned that the study did not require ethical approval, it might be useful to clarify why it was waived and whether any ethical guidelines were followed in retrospective data collection.
Informed Consent: If patient data was used, a brief note on data anonymization and confidentiality would strengthen the ethical clarity.
Recommendations for Corrections:
Refine the abstract to make it more concise and impactful.
Streamline the introduction to avoid redundant information.
Clarify the methodology, particularly how outcomes were measured and how feeding categories were defined.
Improve table readability by reformatting them for clarity.
Ensure consistency in terminology throughout the paper.
Revise grammar and sentence structure for improved readability.
Enhance the ethical considerations section to explicitly mention ethical guidelines followed.
Overall Evaluation:
This is a well-conducted study with meaningful contributions to neonatal care. Addressing the above points will improve the clarity, coherence, and overall impact of the manuscript.
Author Response
Your manuscript has been reviewed by experts in the field and we request that you make major revisions before it is processed further. Please revise your manuscript according to the reviewers' comments and upload the revised file within 10 days. Please click on the "Peer Review Reports" below to find the reviewers' comments and the version of your manuscript to be used for your revisions.
An interesting and fascinating article worth publishing after referring to my comments:
Review of Manuscript: "Impact of Enteral Nutrition on Clinical Outcomes in Very Low Birth Weight Infants in the NICU"
Strengths of the Manuscript:
Relevant Topic & Clinical Importance: The study addresses an important issue in neonatology—feeding strategies for very low birth weight (VLBW) infants—which has significant implications for clinical practice.
Comprehensive Literature Review: The introduction and discussion provide a thorough overview of the current literature, highlighting the importance of human milk in neonatal care.
Statistical Analyses & Adjustments: The manuscript includes robust statistical analyses, adjusting for potential confounders, which strengthens the credibility of the findings.
Clear Findings: The results effectively demonstrate the protective effects of human milk in preventing necrotizing enterocolitis (NEC), late-onset sepsis (LOS), retinopathy of prematurity (ROP), and bronchopulmonary dysplasia (BPD).
Areas for Improvement:
1. Clarity & Structure:
Abstract: The abstract is well-structured but can be slightly more concise. Consider summarizing key numerical results in a clearer manner to improve readability.
Changes have been introduced to improve readability and effectiveness.
Introduction: While informative, some parts of the introduction are redundant. The emphasis on the benefits of human milk is repeated multiple times—consider streamlining these sections to avoid redundancy.
We have eliminated redundancies.
Discussion: While comprehensive, some sections are lengthy and could be better structured into subsections focusing on each neonatal complication (NEC, LOS, ROP, BPD) with more concise interpretations.
We followed the advice and aimed to shorten the longer sentences.
Methodological Considerations:
Study Design Description: The study design could be more clearly described. The inclusion and exclusion criteria should be explicitly stated in a structured format.
We improved the clarity of the study description based on the feedback provided.
Statistical Methods: The statistical section is thorough but could benefit from a brief explanation of why specific tests (e.g., Fisher’s exact test, Spearman’s correlation) were chosen for different variables.
Confounding Factors: While the study adjusts for confounders, it would be useful to elaborate on any potential unmeasured confounders that might have influenced the findings.
Other factors, such as growth retardation or chorioamnionitis, were included in the database but were not assessed as confounding factors. I'm unsure whether we should mention them
Data Presentation & Tables:
Tables & Figures Formatting: Some tables contain dense text, making them difficult to interpret at first glance. Consider using more spacing, highlighting key results, and ensuring consistent formatting across all tables.
Figure Legends: Some figures, such as Figures 1 and 2, should have clearer legends explaining what each comparison represents.
Statistical Significance Indicators: Ensure all p-values and confidence intervals are consistently reported across tables and figures.
We removed the first figure, added explanations to each table and figure, and reviewed all of them.
- Language & Grammar:
Grammar & Sentence Structure: Some sentences are overly complex. Shortening them would improve readability. Consider revising passive voice constructions for a more direct approach.
We shortened sentences that were too long and we tried to eliminate passive voice constructions .
Terminology Consistency: Throughout the paper, ensure that terms such as “Mother’s Own Milk (MOM)” and “Donor Human Milk (DHM)” are used consistently.
Done.
- Ethical Considerations:
Ethical Approval Statement: While it is mentioned that the study did not require ethical approval, it might be useful to clarify why it was waived and whether any ethical guidelines were followed in retrospective data collection.
Informed Consent: If patient data was used, a brief note on data anonymization and confidentiality would strengthen the ethical clarity.
We added a clarification regarding ethical concerns.
Recommendations for Corrections:
Refine the abstract to make it more concise and impactful.
Streamline the introduction to avoid redundant information.
Clarify the methodology, particularly how outcomes were measured and how feeding categories were defined.
Improve table readability by reformatting them for clarity.
Ensure consistency in terminology throughout the paper.
Revise grammar and sentence structure for improved readability.
Enhance the ethical considerations section to explicitly mention ethical guidelines followed.
Overall Evaluation:
This is a well-conducted study with meaningful contributions to neonatal care. Addressing the above points will improve the clarity, coherence, and overall impact of the manuscript.
While we hope to have followed the suggested guidelines and improved the effectiveness and clarity of the text, we are grateful for the feedback provided.
Reviewer 2 Report
Comments and Suggestions for Authors
Be it over a period of about 2 decades, the current paper describes for a single center the expected outcome of association of MOM or human milk with prevention of adverse events during neonatal intensive care. In this way, the current cohort rather confirms the construct on the benefits of human milk or MOM.
There is likely benefit to better quantify the impact, along NNT to prevent an event, of provide at least some more quantitative information in the abstract. Why very reasonable as a hypothesis, associations still are different from causality, so that I do recommend to somewhat adapt the wording.
Over the time interval, other practices like prenatal care, or prenatal lung maturation may also have changed. Have the authors any data to reflect on this ? Furhermore, the cut off was set on weight (VLBW), and not gestational age, so that I expect that SGA is ‘overweighted’ in the current cohort. Can the authors provide some additional analysis in either AGA or SGA cases ? How have you handle non-survivors in this analysis ?
As the analysis is retrospective, and NEX is notoriously poor on interrater variability for Bell stage 1, there is likely value to further reflect on this.
Editing
2.2. statistical analyses: please check as the text flow seems inaccurate.
Not sure if there is real add on value for the figure 1 and figure 2, and the same holds true for figure 3 (mechanical ventilation, instead of meccanic ?) Figure 4 likely needs ‘axes’ description ?
Ethics
I’m not sure if no ethics approval were needed. I would expect that an EC/IRB would waive informed consent because of the retrospective design and analysis, but I would still expect their statement since data were collected and analysed.
Author Response
Be it over a period of about 2 decades, the current paper describes for a single center the expected outcome of association of MOM or human milk with prevention of adverse events during neonatal intensive care. In this way, the current cohort rather confirms the construct on the benefits of human milk or MOM.
There is likely benefit to better quantify the impact, along NNT to prevent an event, of provide at least some more quantitative information in the abstract. Why very reasonable as a hypothesis, associations still are different from causality, so that I do recommend to somewhat adapt the wording.
Added
Over the time interval, other practices like prenatal care, or prenatal lung maturation may also have changed. Have the authors any data to reflect on this ?
We added data on the reduction in the use of invasive ventilation.
Furhermore, the cut off was set on weight (VLBW), and not gestational age, so that I expect that SGA is ‘overweighted’ in the current cohort. Can the authors provide some additional analysis in either AGA or SGA cases ? How have you handle non-survivors in this analysis ?
In addition to the weight criterion, there is also the criterion of gestational age < 32 weeks. SGA (small for gestational age) infants were included among the risk factors.
As the analysis is retrospective, and NEX is notoriously poor on interrater variability for Bell stage 1, there is likely value to further reflect on this.
This has been added both within limits and in the discussion
Editing
2.2. statistical analyses: please check as the text flow seems inaccurate.
Not sure if there is real add on value for the figure 1 and figure 2, and the same holds true for figure 3 (mechanical ventilation, instead of meccanic ?) Figure 4 likely needs ‘axes’ description ?
Figure 1 has been removed. In the others, the specification has been added.
Ethics
I’m not sure if no ethics approval were needed. I would expect that an EC/IRB would waive informed consent because of the retrospective design and analysis, but I would still expect their statement since data were collected and analysed.
We have added the clarification that The study was approved by the hospital’s Technical-Scientific Committee (TSC), which determined that regional ethics committee review was not required, as data collection was conducted in full compliance with anonymization protocols, in accordance with TSC guidelines.
Round 2
Reviewer 2 Report
Comments and Suggestions for Authors
the comments have been handled well
Author Response
It is well known that SGA (small for gestational age) infants have a double risk of developing NEC, and in our endpoints, we do not investigate any differences between SGA and AGA (appropriate for gestational age) due to the small sample size under examination. In our population, SGA infants represent only 15% of the sample, and specifically among the newborns who presented with NEC, only one fell into the SGA category. The purpose of our study is not to focus on the risk factors for NEC but to study the impact of enteral feeding on different outcomes.
The specification of weight in the sample description is related to the definition of VLBW and we have added a gestational age limit of 32 weeks.
Among the confounding factors, we considered everything that could have affected the outcomes, including the type of delivery, characteristics of birth such as the Apgar score, the need for resuscitation, and exposure to corticosteroids. We also evaluated the presence of other comorbidities beyond those examined, but we did not obtain significant assessments. We did not assess the actual availability of human milk for each case, but we studied the feeding trends over the year, specifically the availability of human donor milk.
We have incorporated NNT values into the Results and Discussion sections that have been highlighted in yellow in the manuscript as requested by the reviewer